# Pancreatic Neuroendocrine Neoplasms in Multiple Endocrine Neoplasia Type 1

**DOI:** 10.3390/ijms22084041

**Published:** 2021-04-14

**Authors:** Francesca Marini, Francesca Giusti, Francesco Tonelli, Maria Luisa Brandi

**Affiliations:** 1Department of Experimental and Clinical Biomedical Sciences, University of Florence, Viale Pieraccini 6, 50139 Florence, Italy; francesca.marini@unifi.it (F.M.); francesca.giusti@unifi.it (F.G.); 2F.I.R.M.O. Italian Foundation for the Research on Bone Diseases, Via Reginaldo Giuliani 195/A, 50141 Florence, Italy; francesco.tonelli@unifi.it

**Keywords:** Multiple Endocrine Neoplasia type 1 (MEN1), pancreatic neuroendocrine tumors (pNETs), *MEN1* gene, gene mutation, menin, epigenetic factors

## Abstract

Pancreatic neuroendocrine tumors (pNETs) are a rare group of cancers accounting for about 1–2% of all pancreatic neoplasms. About 10% of pNETs arise within endocrine tumor syndromes, such as Multiple Endocrine Neoplasia type 1 (MEN1). pNETs affect 30–80% of MEN1 patients, manifesting prevalently as multiple microadenomas. pNETs in patients with MEN1 are particularly difficult to treat due to differences in their growth potential, their multiplicity, the frequent requirement of extensive surgery, the high rate of post-operative recurrences, and the concomitant development of other tumors. MEN1 syndrome is caused by germinal heterozygote inactivating mutation of the *MEN1* gene, encoding the menin tumor suppressor protein. MEN1-related pNETs develop following the complete loss of function of wild-type menin. Menin is a key regulator of endocrine cell plasticity and its loss in these cells is sufficient for tumor initiation. Somatic biallelic loss of wild-type menin in the neuroendocrine pancreas presumably alters the epigenetic control of gene expression, mediated by histone modifications and DNA hypermethylation, as a driver of MEN1-associated pNET tumorigenesis. In this light, epigenetic-based therapies aimed to correct the altered DNA methylation, and/or histone modifications might be a possible therapeutic strategy for MEN1 pNETs, for whom standard treatments fail.

## 1. Pancreatic Neuroendocrine Tumors: Classification and Pathology

Pancreatic neuroendocrine tumors (pNETs) are rare primary cancers of the endocrine pancreas accounting less than 3% of all pancreatic neoplasms. pNETs manifest mainly as sporadic tumors, or arise in the context of genetically determined syndromes in less than 10% of cases.

pNETs are primarily distinguished between functioning (15%) and nonfunctioning tumors (85%). Functioning pNETs retain secretory ability and release excessive amounts of hormones (i.e., insulin, gastrin, glucagon), which cause specific endocrine syndromes and related symptoms. Conversely, nonfunctioning pNETs (NF-pNETs) are more poorly differentiated tumors which do not secrete any hormone or active peptide, are often asymptomatic, and are discovered incidentally through abdominal screening performed for other reasons, or if their growth causes a symptom-inducing compression of adjacent structures (i.e., obstruction of the pancreas/bile duct), usually when they have already metastasized the adjacent lymph nodes and/or the liver. NF-pNETs have a worse prognosis; patients present a high percentage of unresectable disease, often with liver metastasis, and a five-year survival rate of 30–40% [1].

According to the 2017 WHO classification for pNETs [2] and the 2019 WHO classification of tumors of the digestive system [3], pNETs and pancreatic neuroendocrine carcinomas (pNECs) are included in the superfamily of pancreatic neuroendocrine neoplasms (pNENs). Pathological classification of all pNENs includes tumor grade and malignity; pNENs can be ascribed three different grades based on tumor cell proliferation rate, determined by mitotic count and/or Ki-67 nuclear staining index [2]. Grade 1 (G1) includes well-differentiated low-grade pNETs with a Ki-67 index less than 3% of ≥500 cells and/or a mitotic count less than 2 per 10 high-power fields. Grade 2 (G2) are well-differentiated intermediate-grade pNETs having a Ki-67 index of 3–20% of ≥500 cells and/or a mitotic count of 2–20 per 10 high-power fields. Grade 3 (G3) includes high-grade tumors presenting a Ki-67 index over 20% of ≥500 cells and/or a mitotic count over 20 per 10 high-power fields. Inclusion in the G3 group is based only on the values of Ki-67 index and/or mitotic count, regardless of the other morphological features of tumors. G3 contains two subgroups of cancers which, despite their similar proliferation rate, are different for both their malignancy degree and their genetic background: (1) well-differentiated high-grade pNETs with round homogeneous nuclei, low-to-moderate atypia, salt and pepper-like chromatin, and fine granular cytoplasm, with frequent abnormalities of the *MEN1* tumor suppressor gene (25–44%), the apoptotic regulator Death Domain-Associated Protein (*DAXX*) gene (25%), or the chromatin modifier Alpha-Thalassemia/mental Retardation X-linked (*ATRX*) gene (18%); (2) poorly-differentiated high-grade pNECs (small cell type and large cell type), which usually have an extremely high mitotic rate (over 50%) and are commonly mutated on the Retinoblastoma (*RB*) or the Tumoral Protein 53 (*TP53*) tumor suppressor genes. The different genetic bases demonstrate that G3-pNECs do not derive from a malignant progression of well-differentiated pNETs but develop as separate tumors with a more severe grade of malignancy.

Investigation of the molecular pathogenesis of pNETs is limited by their rarity, multiple oncological and endocrinological outcomes, and heterogeneous genetics.

## 2. pNETs in Multiple Endocrine Neoplasia Type 1

pNETs arise in 30–80% of patients with Multiple Endocrine Neoplasia type 1 (MEN1), a rare inherited cancer syndrome characterized by the development of multiple neuroendocrine and nonendocrine tumors in a single patient and caused by a germline heterozygote loss-of-function mutation of the *MEN1* tumor suppressor gene.

A great majority of MEN1 patients develop, during their life, multiple pancreatic microadenomas (less than 0.5 cm in diameter), mostly NF-pNETs (approximately 50% of cases) [4], insulinomas (7–31% of cases) [5], and gastrinomas (about 5% of cases) [6]. Solitary pNETs in MEN1 manifest less frequently (less than 13% of cases), usually as large masses (macroadenomas > 2 cm), and are often nonfunctional [7]. MEN1 pNETs manifest at a younger mean age compared to their sporadic counterparts (10–50 years vs. 50–80 years). MEN1 gastrinomas are prevalently located within the duodenum (over 90% of gastrinomas), and less than 10% of cases affect the pancreas. Glucagonomas, somatostatinomas and vasoactive intestinal polypeptide secreting tumors (VIPomas) manifest less frequently (3–4%, <2% and <2%, respectively) [5]. Neuroendocrine tumors affecting the gastro-entero-pancreatic tract in MEN1 syndrome are shown in Figure 1.

Secreting pNETs are usually associated with distinct endocrine syndromes, caused by the excessive amount of hormone released by tumor cells. As their sporadic counterpart, MEN1 pNETs result positive to immunostaining for neuroendocrine markers and, in case of functioning tumors, their specific secreted hormones.

MEN1 pNETs follow the same three-grade WHO classification and they usually present a slow growth rate, with an estimated doubling time of 5–10 years [9]. However, in some cases, MEN1-associate pancreatic tumors exhibit a more aggressive behavior than the sporadic forms, with metastasis described in about 23–33% of cases. Tumor size is an important predictive factor of prognosis and overall survival in MEN1 pNETs, mostly for NF-pNETs; tumors larger than 2 cm in diameter have been associated with a high risk of malignancy and metastases development. Even if characterized by a slow disease progression, when compared with the non-endocrine pancreatic cancers and their endocrine sporadic counterpart, and despite the availability of an early genetic and clinical diagnosis and the possibility of praecox surgical intervention, pNETs currently remain the main cause of death among MEN1 patients, with an estimated 10-year survival after diagnosis of NF-pNETs ranging from 23% to 62% [10].

Currently, no prognostic factors allow clear identification of MEN1 patients at risk of metastatic cancers, nor to tailor personalized pNET treatment. KI-67 labeling and mitotic count have both been shown to be prognostic in sporadic pNETs [11,12], but few data are available for MEN1 pNETs. A study by Conemans et al. [13] evaluated, in formalin-fixed-paraffin-embedded pNET tissues from 69 MEN1 patients, the prognostic value of Ki-67 labeling index and mitotic rate, according to the WHO grade, with respect to clinical features of the tumors (i.e., tumor size, presence/absence of liver metastases, and time of metastases diagnosis after identification of the primary tumor). They showed that the mitotic count value was associated with the development of metastases only for NF-pNETs, but not for insulinomas, with 80% of metastases occurring in the G2 cancers (mitotic rate 2–20 per 10 high-power fields) and only in 23% of G1 tumors (mitotic rate <2 per 10 high-power fields). No significant association was found between values of the Ki-67 labeling index and metastases development. Moreover, the authors confirmed that tumor size is the main risk factor associated with liver metastases, both for insulinomas and NF-pNETs (in their cohort, the cut off diameter of primary tumor causing metastases was 3 cm, with respect to the 2 cm previously indicated by the guidelines) [13].

Unfortunately, it is not possible to predict cancer type development, tumor behavior, or risk of metastases based on the *MEN1* gene mutation. MEN1 patients have a variable clinical expressivity; clinical manifestations, including pNETs, can be different between members of the same family bearing the same *MEN1* mutation, and even between homozygous twins, suggesting that other genetic, epigenetic, and environmental factors can affect MEN1 tumorigenesis and drive individual tumor development and progression.

## 3. Genetics and Epigenetics of MEN1 pNETs

The exact in vivo mechanisms by which *MEN1* mutations initiate and drive endocrine pancreas tumorigenesis have not yet been fully explained. *Men1*-mutated mouse models have yielded important insights in MEN1-related pancreatic tumorigenesis. Mice bearing a germline heterozygous inactivation of the *Men1* gene are a representative model of MEN1 in humans; these animals developed hyperplasia and/or tumors of the pancreatic beta cells (mainly insulinoma), starting at 8 to 12 months of life [14]. Tumor lesions are characterized by somatic loss of heterozygosity (LOH) of the *Men1* locus. Somatic genetic analyses of the pancreas have shown that pancreatic islets manifest a pretumor hyperplastic stage in which *Men1* LOH has not yet occurred [15]. Mice with the conditional homozygous knockout of the *Men1* gene in the whole pancreas developed tumors derived from the neuroendocrine beta cells; no tumors of the exocrine pancreas manifested [16]. Mice with the conditional homozygous knockout of the *Men1* gene in the alpha cells of the endocrine pancreas did not develop glucagonomas, but they prevalently developed tumors of the beta cells, mainly insulinomas, suggesting that loss of *Men1* functions could induce the trans-differentiation of alpha cells to beta cells, or condition alpha cells to release paracrine signals that lead to increased proliferation of beta cells [17,18].

In MEN1 patients, inherited germline heterozygote inactivating mutations of the *MEN1* tumor suppressor gene are responsible for the development of pancreatic multiple adenomas. The first germline mutation is inherited from the affected parent or, very rarely, developed de novo at the embryonic level. The loss of the second copy of the gene, by LOH at the 11q13 locus (over 90% of cases) or, more rarely (about 10% of cases), by the development of an intragenic loss-of-function mutation, happens at the somatic level in neuroendocrine cells, and drives tumor development and progression [19]. Menin, the protein encoded by the *MEN1* gene, is a key regulator of endocrine cell plasticity, and its loss in these cells is sufficient for tumor initiation.

The *MEN1* gene has been found to be the most common somatically mutated gene in sporadic pNETs (44% of total cases, 30% of NF-pNETs, 7% of insulinomas, 36% of gastrinomas, 67% of glucagonomas, and 44% of VIPomas) [9,16], and LOH at 11q13 or somatic *MEN1* mutations manifest in up to 46% of these tumors (about 39% of NF-pNETs and 15–20% of gastrinomas and insulinomas), independent of the tumor stage [20,21,22]. Moreover, subcellular localization of menin is altered in about 80% of sporadic pNETs [23]. All these data confirm the key role that loss of function of menin has in neuroendocrine pancreas tumorigenesis, both for sporadic and syndromic inherited forms.

Menin is a ubiquitously expressed protein of 610 amino acids, primarily located in the nucleus, thanks to the presence of three nuclear localization signals (NLSs) in the C-terminal region. The nuclear localization is fundamental to exert its biological and antitumor functions, such as regulation of DNA replication and repair, histone methylation and acetylation, DNA methylation, positive or negative control of gene expression, cell signaling, control of cell cycle and cell growth, control of apoptosis, and regulation of cell mobility and adhesion. Most *MEN1* mutations, in sporadic and syndromic pNETs, appear to alter/remove the region of the gene involved in the nuclear localization. Immunostaining of menin in tumor specimens has shown a low-to-absent presence of the protein in the nucleus, compared to normal tissue [23]. Menin has been shown to be located mainly in the nucleus of nongrowing cells or, conversely, in the cytoplasm in dividing cells.

This protein lacks any sequence and/or motif similarity to any known proteins and this makes it difficult to understand its mechanisms of activity as a tumor suppressor gene. To date, direct or indirect interactions have been demonstrated with over 50 different protein partners of known functions, mostly transcription factors and epigenetic regulators.

The absence of a clear genotype-phenotype association in MEN1 syndrome reinforces the hypothesis that the *MEN1* mutation is responsible for the penetrance and development of the syndrome; however, other cofactors, such as epigenetic mechanisms and environmental influences, could drive individual tumorigenesis, even in presence of the same *MEN1* mutation. Alteration of epigenetic mechanisms, consequent to loss of menin activity, which modify gene expression, seems to have a key role in development and progression of MEN1 pNETs.

Epigenetic mechanisms identified to date that are altered by menin loss are reported in Table 1.

One of the menin interacting partners is DAXX, a pleiotropic protein acting as a transcriptional repressor, being a component of the chromatin remodeling complex and involved in the regulation of apoptosis and maintenance of the stable telomere length. Since *MEN1*, *DAXX*, and *ATRX*, which were all found to be mutated in a large percentage of pNETs, are all involved in the control of the correct chromatin remodeling, it can be hypothesized that the pNET tumorigenesis, both in sporadic and MEN1-related forms, could be regulated by an altered control of gene expression via this epigenetic mechanism. Mutations in *DAXX* or *ATRX* have been associated, in sporadic pNETs, with alternative lengthening of telomeres (ALT), a telomerase-independent process that is activated in tumor cells to prevent telomere shortening during cell proliferation and avoid senescence of proliferating cells [24]. In a study by Cejas et al. [24], ALT was found in 48% of sporadic and 25% of MEN1 NF-pNETs, also resulting in it being significantly associated with postoperative distant recurrences. However, although the assessment of ALT status could offer useful postoperative prognostic information for pNETs, the complexity of its laboratory analysis has prevented it from becoming a routine clinical test to tailor postoperative clinical decisions.

Normally, menin and DAXX proteins directly interact with each other and crosstalk to suppress the proliferation of neuroendocrine cells by inhibiting expression of the Membrane Metallo-Endopeptidase (MME). Both menin and DAXX are required to activate the SUV39H1 methyltransferase, which induces the H3K9me3, repressing transcription of target genes associated with growth of neuroendocrine cells, such as *MME*, *GBX2* and *IL6*. The knockout of *Men1* or *Daxx* in mouse embryonic fibroblasts resulted in an increased expression of *Mme, DCN* and *Gria3* genes [25]. Increased expression of MME has been found in numerous solid cancers, including neuroendocrine tumors, and it has been correlated with large tumor size, a high proliferation rate and presence of metastases, as well as with a higher risk of liver metastasis from colon cancer [25]. Therefore, restoring the correct function of SUV39H1, repressing *MME* expression, or directly inhibiting MME protein, may be effective approaches for treatment of pNETs missing function of menin, DAXX, or both. Treatment of the rat insulinoma cell line, INS-1, knocked out for *Men1* or *Daxx*, with Thiorphan, an MME inhibitor, was shown to repress cell proliferation at different time points in a dose-dependent manner [25], indicating this small molecule as a promising drug for the control of tumor growth in pNETs with *MEN1* and/or *DAXX* mutations.

Gene expression or silencing are both epigenetically regulated by the level of methylation/demethylation of lysine residues of histone 3 (H3). The trimethylation of H3 on the lysine 4 residue (H3K4me3) is a common epigenetic mark of actively transcribed genes, while trimethylation on lysine 9 and 27 residues (H3K9me3 and H3K27me3) are both associated with transcriptional inhibition. Wild-type menin has been shown to induce the H3K4me3, presumably via its direct interaction with Mixed Lineage Leukemia proteins (MLL1/KMT2A and MLL2/KMT2B), two histone lysine methyltransferases (KMTs) that possess intrinsic H3K4 methyltransferase activity [26]. To elucidate the role of H3K4me3, mediated by the menin-methyltransferase complexes, during the pNET tumorigenesis, Lin et al. [27] evaluated, in vivo, the H3K4me3 and H3K27me3 signals in the beta-islets of 2-month-old mice bearing a germinal conditional homozygous knockout of the *Men1* gene in the pancreas beta cells, with respect to control RIP-Cre mice. The integrative analysis of H3K4me3 and H3K27me3 levels, and of gene expression profiles via the unbiased chromatin immunoprecipitation, coupled with next-generation sequencing, showed that the absence of menin correlated with the loss of H3K4me3 and a mirrored increase of H3K27me3, at promoters of a specific set of target genes, which resulted in downregulation of gene expression. Many of these target genes were linked to type 2 diabetes and beta cell functions, with the Insulin -like Growth Factor 2 mRNA Binding Protein 2 (*Igf2bp2*) gene, encoding the insulin-like growth factor 2 mRNA binding protein 2, being one of the most downregulated by the H3K4me3 loss. Moreover, Lin et al. [27] evaluated in mice how H3K4me3 and H3K27me3 signals changed over time, at 2, 6, and 12 months of age, at the promoter of the *Igf2bp2* during the *Men1*-knockout-driven beta-islet tumorigenesis. The H3K4me3 signal was reduced at 2 months of age in *Men1*-deficient beta islets with respect to those of control mice. At 6 and 12 months, H3K4me3 signal was reduced with respect to 2 months, both for *Men1* mutated and control islets. Conversely, H3K27me3 showed an inverse occupancy of *Igf2bp2* promoter over time, with no detectable signal at 2 months in *Men1* mutated beta islets and a strong and progressive increase at 6 and 12 months. Other gene promoters (i.e., *Gata6* and *Otxr*) showed a similar inverse pattern of H3K4me3 and H3K27me3 signals during the progression of beta islet tumorigenesis.

These altered epigenetic mechanisms (loss of H3K4me3 and gain of H3K27me3), subsequent to menin absence/reduction, which appear to drive the MEN1-associated beta cell tumorigenesis, can be reversed by inhibiting/silencing the H3K4me3 demethylase Rbp2, suggesting this enzyme as a possible target of an early-stage epigenetic therapy for tumors of the beta cells in MEN1 patients [27].

Normally, menin suppresses Hedgehog (Hh) signaling, a pro-proliferative pathway, via its direct interaction and activation of the protein arginine methyltransferase 5 (PRMT5), which, thus, methylases histone 4 on the arginine 3 residue (H4R3me2), repressing gene expression at the promoter of the Growth Arrest Specific 1 (*GAS1*) gene [28]. GAS1 is a key factor for the binding of the Sonic hedgehog (Shh) ligand to its receptor PTCH1, a 12-pass transmembrane protein that is the major receptor for mammalian Hh signals. When Shh binds PTCH1, the receptor loses its suppression activity on the transmembrane protein Smoothened (SMO), resulting in the activation of Hh signaling and subsequent transcription of target genes, including pro-proliferative genes [29]. Gurung et al. [30] demonstrated, in murine MEN1 pNETs of beta islets, that the loss of menin function results in an increased expression of *Gas1* and *Ptch1,* and enhances the SMO-mediated activation of Hh signaling, leading to tumor cell growth. In the same study [30], the authors tested the effect of a 4-week treatment with the Hh signaling inhibitor GDC-0449 (SMO inhibitor) on 8-month-old *Men1*-mutated mice with insulinomas, showing a reduction of proliferation of beta cells of approximately 60% and a reduction of circulating levels of insulin. At molecular level, treatment with GDC-0449 induced, in the beta islets, a reduction of *Ptch1* expression, with respect to *Men1* mutated mice treated with a control vehicle. In conclusion, GDC-0449 was shown to effectively suppress both proliferation of MEN1 tumor beta cells in vivo and reduce insulin secretion by insulinomas, revealing itself to be a promising therapy against MEN1 insulin-secreting pNETs. In general, the menin/PRMT5/Hedgehog signaling pathway and its molecular actors are potential targets for the treatment of MEN1 pNETs [31].

These menin-regulated pathways with potential role in pNET tumorigenesis are depicted in Figure 2.

In addition, since *MEN1* inactivating mutations are known to affect DNA methylation, a significant role of this epigenetic mechanism is suggested in the development and progression of tumors in *MEN1*-mutated neuroendocrine cells of the pancreas, both in the syndromic form and in sporadic pNETs with somatic biallelic inactivation of the *MEN1* gene. The reversible methylation of CpG islands in promoter/enhancer regions is an epigenetic dynamic mechanism, controlled by methylating and demethylating enzymes, commonly regulating gene expression. The hypermethylation of promoters, which silences the expression of tumor suppressor genes, and the genome-wide hypomethylation, which leads to DNA instability, are common hallmarks of cancer. Whole-genome analysis of DNA methylation has shown the presence of hypermethylation in MEN1-associated pNETs with respect to VHL-mutated and sporadic pNETs [32], in association with a significantly higher number of hypermethylated CpG islands and downregulated genes. Interestingly, pNETs with *ATRX*, *DAXX*, or *MEN1* mutations showed common DNA methylation signatures, similar to those of the alpha cells of the pancreas, and different from pNETs without mutations of these three genes [33].

A study by Conemans et al. [34] evaluated the DNA methylation profile in the promoters of 56 tumor suppressor genes between 61 formalin-fixed-paraffin-embedded MEN1 pNET specimens and 34 sporadic pNET tissues, and between different subgroups of MEN1 pNETs. The hypermethylation of CpG sites was confirmed to be a common event, both in sporadic and MEN1 tumors. The cumulative methylation index (CMI) showed no significant difference between MEN1 and sporadic pNETs, but it resulted in significantly higher, among MEN1 NF-pNETs, tumors larger than 2 cm and liver metastatic cancers. This result suggests CMI as a possible prognostic factor for malignancy in MEN1 NF-pNETs, a direct role of this epigenetic modification in the malignant progression of these tumors, and that hypermethylation of promoters could be a therapeutic target in MEN1 patients with advanced NF-pNETs. The analysis of frequency and level of promoter methylation of single genes resulted: (1) in significantly more frequent and higher for the caspase 8 (*CASP8*) gene in MEN1 pNETs than in the sporadic counterpart, and higher in MEN1 pNETs and sporadic pNETs negative for menin staining with respect to sporadic pNETs retaining wild-type menin expression; (2) in significantly higher for the Ras Association Domain Family Member 1 (*RASSF1*) tumor suppressor gene in MEN1 insulinomas than in MEN1 NF-pNETs; and (3) in significantly more frequent for the O-6-Methylguanine-DNA Methyltransferase (*MGMT*) gene in MEN1 NF-pNETs than in MEN1 insulinomas.

Hypermethylation of *CASP8* promoter appears to be a hallmark of menin loss in pNETs. Wild-type menin induces the H3K4me3, which has been shown to counteract the activity of DNA methyltransferase by protecting the CpG islands from DNA methylation, thus, preventing the methylation of the *CASP8* promoter. The *CASP8* gene encodes the caspase 8 protein, which activates apoptosis; the hypermethylation of *CASP8* promoter, consequent to the loss of menin function and H3K4me3, silences the expression of the gene and may have a role in development and progression of *MEN1*-mutated pNETs by preventing apoptosis of tumor cells, as previously described in other human tumor types [35]. In this light, the block of methylation of *CASP8* promoter or, alternatively, the restoration of H3K4me3, could be possible epigenetic therapeutic approaches in pNETs with homozygote *MEN1* mutations.

The promoter of *RASSF1,* a tumor suppressor gene involved in induction of cell cycle arrest, has been reported to be hypermethylated, and inactivated, in 63–83% of sporadic pNETs (mostly NF-pNETs), and in 99% and 97% of MEN1 pNETs for *RASSF1_2* and *RASSF1_1*, respectively [34].

The hypermethylation of *MGMT* promoter has been described in about 40% of sporadic pNETs, 44.7% of MEN1 NF-pNETs, and only 8.3% of MEN1 insulinomas [34]. *MGMT* gene encodes the O-6-methylguanine-DNA-methyltransferase, a DNA repairing enzyme that removes the guanine-alkyl group added by alkylating agents, such as temozolomide, providing cells with defense against mutagenesis and toxicity from alkylating agents. Loss of O-6-methylguanine-DNA-methyltransferase activity may favor the carcinogenetic action of exogenous alkylating agents and promote tumor development. On the other hand, the presence of *MGMT* promoter hypermethylation in a percentage of MEN1 NF-pNETs could increase their sensitivity and responsiveness to treatment with temozolomide and other alkylating agents; evaluation of methylation status of the *MGMT* promoter in tumor biopsies may serve as a predictive pharmacoepigenetic biomarker of response to chemotherapeutic alkylating molecules in MEN1 patients with advanced NF-pNETs.

Hypermethylation of promoters of *RASSF1* and *MGMT* genes appears to be an epigenetic marker of MEN1 insulinomas or MEN1 NF-pNETs, respectively, suggesting that they could be promising selective therapeutic targets for these two types of pancreatic tumors. DNA hypermethylation, associated with pNET tumorigenesis, could be blocked by directly inhibiting DNA methyltransferases (DNMTs).

Noncoding RNAs (ncRNAs) are a class of epigenetic factors regulating gene expression, whose role has been investigated in MEN1 tumorigenesis. They include long noncoding RNAs (over 200 nucleotides in length) and short noncoding RNAs (less than 30 nucleotides). Among the short noncoding RNAs, microRNAs (miRNAs), a class of negative regulators of gene expression at the post-transcriptional level, are important actors in the regulation of numerous biological processes, and the altered expression/activity of some of them has been associated with the development of various human malignancies. Vijayaraghavan et al. [36] showed, in vitro, on MIN6 mouse insulinoma cells and βlox5 immortalized human beta-cell lines, that the miR-24 directly decreases menin levels via a feedback loop between *MEN1*-miR-24-menin, similar to that previously shown in parathyroid tissues from MEN1 patients [37]. This mechanism of menin epigenetic silencing is suspected to mimic the “second somatic hit” of *MEN1* inactivation and trigger tumor development in neuroendocrine cells not yet manifesting the LOH at the *MEN1* locus [37]. As a result of increased miR-24 levels, both the pancreas cell lines showed a reduction of menin, p27^kip1^ and p18^INK4C^ expression [36], suggesting that, in the process of MEN1 pNET tumorigenesis, miR-24 may act as a pro-oncogenic factor (oncomir) by directly silencing menin and indirectly promoting cell division. In this light, targeting miR-24 by a specific antisense RNA (antagomir) could restore the normal expression/activity of wild-type menin in pancreas neuroendocrine cells of MEN1 patients still retaining a wild-type copy of the *MEN1* gene.

## 4. Therapy of MEN1 pNETs: Current Approaches and Future Perspectives

Surgical resection of MEN1 pNETs is employed to alleviate the endocrine syndromes caused by functioning pNETs which cannot be controlled by specific medical therapy, and/or to reduce the risk of malignant progression and metastases. Timing and extent of surgery for MEN1 pNETs are still debated, and they vary according to the different types of pNETs to be treated. Gastrinomas are localized almost exclusively in the duodenum, often multiple and of small or microscopic dimensions, so that they can rarely be successfully treated by enucleation or enucleoresection of the duodenal mucosa. They usually require aggressive surgery for resection of the duodenum and part of the pancreas. Surgery for gastrinomas is suggested in the presence of concomitant NF-pNET close to 2 cm in diameter and/or presenting a doubling time of six months or less [7]. Surgery for insulinoma can range from single tumor enucleation to limited pancreas resection, to resection of half of the pancreas together with enucleation of remaining smaller tumors, depending both on size and numbers of tumors [7]. Surgery for NF-pNETs is mainly guided by tumor size, which has been directly associated with the risk of metastases; the suggested dimension cut-off ranges from 1 to 3 cm [7].

As opposed to sporadic pNETs, the surgical context in MEN1 is complicated by the frequent presence, in a single patient, of multiple tumors of different kinds and dimensions, scattered through endocrine areas of the pancreas and often requiring extensive surgery with the risk of severe postoperative consequences, such as exocrine gland insufficiency and diabetes. The multiplicity of microadenomas usually prevents surgery from achieving the total removal of all tumors and the complete resolution of the disease. Moreover, given the genetic basis of the syndrome, incidence of post-intervention recurrence is common, and more than one surgery can be required during patient’s lifetime.

Medical therapies for MEN1 pNETs are the same as those for pNETs in non-MEN1 patients, despite a lack of formal evaluation in wide cohorts of MEN1 patients, and their effectiveness has been shown only in single case reports or small case series. These therapies aim to control hormone hypersecretion and/or tumor growth, and are classified into pharmacological treatments, which include biotherapies and chemotherapies, and tumor targeted radiotherapy through the peptide receptor radionuclide therapy (PRRT).

Control of gastrinoma-induced hypergastrinemia is performed with antiacid pharmacological therapy, consisting in the administration, alone or in combination, of proton pomp inhibitors (lansoprazole, omeprazole, pantoprazole) and histamine-2 receptor antagonists (cimetidine, famotidine, ranitidine). This antiacid medical therapy also reduces the risk of development and worsening of type 2 neuroendocrine tumors of the enterochromaffin-like cells of the mucosa or submucosa of the stomach.

Biotherapies for pNETs include hormonal treatment with somatostatin analogues (SSAs), and therapies targeted to tumor-specific altered signaling pathways, such as inhibitors of mammalian target of rapamycin (mTOR) and inhibitors of tyrosine kinase receptors (TKRs).

Somatostatin is a hormone able to inhibit the release of other hormones, cell proliferation and angiogenesis. As their sporadic counterpart, MEN1 pNETs overexpress somatostatin receptors (SSTRs), mainly the subtypes 2 and 5 (SSTR2 and SSTR5); gastrinomas, glucagonomas, VIP-secreting tumors and NF-pNETs overexpress SSTR2 in 80–100% of cases, and insulinomas in about 50–70% of cases [38]. This molecular characteristic, which distinguishes pNETs from healthy pancreas, enables the use of SSAs for both the control of hormone hypersecretion in functioning pNETs and the reduction of cancer growth both for secreting and non-secreting tumors. Long-lasting synthetic SSAs (octreotide and lanreotide) have high affinity for SSTR2, and they have shown long-term safety and effect in reducing pancreatic hormone hyper-secretion (glucagon, gastrin, and VIP), controlling tumor growth in low-grade pNETs (Ki-67 < 5%) [39], and reducing both size and number of type 2 gastric neuroendocrine tumors after six months of treatment, leading to their complete disappearance after one year [40], while they failed to control insulin over-production in a great majority of MEN1 insulinomas [38].

A double-blind, placebo-controlled randomized phase IIIb study (PROMID) [41] showed that treatment with long-acting octreotide had a good anti-proliferative effect—able to double the time to tumor progression—on both functioning and nonfunctioning tumors in patients with well-differentiated metastatic midgut neuroendocrine tumors. The CLARINET study [42], a randomized, double-blind, placebo-controlled, multinational phase III trial, evidenced safety and efficacy of treatment with the long-acting lanreotide autogel in patients with advanced, well-differentiated or moderately differentiated, nonfunctioning, somatostatin receptor-positive metastatic neuroendocrine tumors of grade 1 or 2 (Ki-67 < 10%). The lanreotide-treated group had a significantly more prolonged progression free survival (PFS) than placebo (18 months vs. median not reached). The estimated rates of PFS at 24 months were 65.1% and 33% in the lanreotide or placebo arms, respectively. Neither of these studies included MEN1 patients.

A retrospective study evaluated the therapeutic effect of 12 to 15-month treatment with long-acting octreotide in 40 MEN1 patients with early-stage duodenum-pancreatic neuroendocrine tumors, reporting a tumor size decrease in 10% of cases, the permanence of a stable disease in 80%, and the progression of disease in 10% of patients [43]. Moreover, in six patients presenting abnormally increased serum levels of gastrin, insulin or chromogranin A, this treatment induced a significant clinical and hormonal response in 100% of cases, remaining stable over time [43].

mTOR signaling commonly regulates cell proliferation; somatic mutations in genes involved in this regulative pathway have been identified in about 15% of pNETs. The mTOR oral inhibitor everolimus increased PFS from 6 to 11 months in patients with advanced neuroendocrine tumors, including pNETs [38]. However, pNETs from hereditary syndromes, such as MEN1, were under-represented in pivotal phase III trials on everolimus. Whether *MEN1* mutations may be associated with any differential responses to mTOR inhibitor therapy remains to be definitively established. A recent study [44] assessed the efficacy of everolimus, in terms of PFS and time to treatment failure (TTF), in a cohort of 31 patients with advanced pNETs, including 6 cases with germline mutations of the *MEN1* gene. Despite the low number of MEN1 patients included in the study, the everolimus treatment showed a higher disease control rate in MEN1 pNETs than in the sporadic counterpart (87.5% vs. 68.4%). Moreover, both PFS and TTF resulted higher, in a median follow-up of 26 months, in patients with germline mutations (including two *VHL*-mutated cases) compared to those with sporadic pNETs (median TTF: 16.1 vs. 9.9 months, *p* = 0.888; median PFS: 33.1 vs. 12.3 months, *p* = 0.383).

pNETs are highly vascular with respect to healthy pancreas, and they frequently express TKRs, such as platelet-derived growth factor receptors (PDGFRs), receptors of the vascular endothelial growth factor (VEGFRs), and insulin-like growth factor 1 receptor (IGF1R). Continuous daily administration of sunitinib, a TKR inhibitor antibody that targets VEGFRs and PDGFRs and blocks angiogenesis, was demonstrated in a phase III study to increase the PFS to 11.4 months, compared to the 5.5 months of the placebo, and to improve both the overall survival and the tumor objective response rate (ORR) in patients with advanced pNETs [45]. The study population included only 2 MEN1 pNETs, of a total of 171 cases, and both were assigned to the placebo group, making the evidence of efficacy of sunitinib in MEN1 patients inconclusive. A subsequent, open-label, phase IV trial was performed to provide additional information on this sunitinib-treated population [46], confirming this drug as an efficacious and safe treatment option in patients with advanced/metastatic, well-differentiated, unresectable pNETs, as previously observed in the pivotal phase III trial [46]. Recently, the authors combined results of phases III and IV of the sunitinib treatment trial, confirming the efficacy of this TKR inhibitor in the tumor ORR (16.7%), and in improving both the median PFS (12.9 months) and the median overall survival (54.1 months) [47]. Efficacy and safety of sunitinib treatment for this type of pNET were both confirmed in an open-label, phase II trial on 12 Japanese patients [48]. Unfortunately, data on sunitinib treatment of MEN1 pNETs are still missing.

Another RTK antibody inhibitor, the pazopanib, was shown in a phase II trial to increase tumor ORR and disease control rate by about 20% and over 75%, respectively, in non-MEN1 patients with metastatic gastro-entero-pancreatic neuroendocrine tumors [49].

Chemotherapy in pNETs is administered in case of metastatic tumors, tumors with high proliferative index (Ki-67 > 5% or mitosis over 5/10 per high powered field), tumors presenting a rapid progression, and/or functioning tumors with endocrine symptoms not controlled by biotherapy. Chemotherapy compounds can be classified in five groups, all of them used for the treatment of pNETs: (1) alkylating agents (streptozocin, temozolomide, and cisplatin) which covalently bind to DNA thanks to their alkylating groups, disrupt DNA replication, and induce apoptosis; (2) anti-microtubule agents (etoposide and docetaxel) that disrupt microtubules, preventing the formation of the mitotic spindle required for cell division and blocking cell replication; (3) topoisomerase inhibitors (doxorubicin and irinotecan) that block the activity of topoisomerase enzymes preventing the normal unwinding of DNA that is required during DNA replication or mRNA transcription, and blocking cell growth and gene transcription; (4) antimetabolites [5fluorouracil (5FU) and its prodrugs capecitabine, and gemcitabine], which block enzymes required for DNA synthesis or damage DNA by being directly incorporated, leading to cell apoptosis; (5) cytotoxic antibiotics (actinomycin D, mitomycin C, doxorubicin, and mixoxantrone), which can alkylate DNA or be intercalated into DNA, or generate highly reactive free radicals that damage intracellular molecules, leading, in all cases, to cell apoptosis. A combination of more than one chemotherapy compound showed higher pNET ORR than single drug therapy. No specific trials on MEN1 pNETs are available.

The overexpression of SSTRs is a characteristic of pNET tumor cells that enables the targeted PRRT to selectively deliver cytotoxic doses of a radioactive isotope to the tumor, using SSAs (octreotide, octreotate, dototate, and dotatoc) bound to a α- or β-emitting radioisotope (i.e., ^90^Yttrium or ^177^Lutetium). After the bound of the radioactive SSA to SSTR and the internalization of the SSA-SSTR complex, the ionizing radiation is released within the cell, causing damage to DNA and inducing cell death. PRRT is usually well tolerated, and the acute side effects are commonly mild. PRRT with ^177^Lutetium- or ^90^Yttrium-octreotide has been evaluated in patients with gastro-entero-pancreatic neuroendocrine tumors, reporting a tumor ORR of about 20–60%, a PFS of 20–34 months, and an overall survival of 53 months [39]. Combination of more than one radionuclide and combination of PRRT with chemotherapy showed an increase in tumor response in patients with advanced neuroendocrine tumors (including pNETs): the combination of ^177^Lutetium- and ^90^Yttrium increased overall survival [50] and the combination of ^177^Lutetium-octreotate, capecitabine and temozolomide resulted in a complete or partial response in >50% of the patients [51].

Main currently employed therapies for MEN1 pNETs are classified and summarized in Table 2.

Epigenetic changes appear to be a hallmark of MEN1 pNET development and progression; they are reversible modifications, due to altered activity of specific enzymes, that can be pharmacologically targeted to restore normal epigenetic states, blocking pro-oncogenic pathways and controlling tumor cell growth, hormone over-secretion, and tumor cell dissemination.

A study by Lines et al. [52] examined, in vitro, the pharmacological effects of 9 inhibitors that target different components of epigenetic pathways, 5 targeting histone methylation pathways (UNC0638, UNC0642, SGC0946, IOX-1 and UNC1215), and 4 targeting histone acetylation pathways (JQ1, PFI-1, RVX-280 and C464), on the BON1 human pNET-derived cell line. JQ1, PFI-1, and IOX-1 reduced BON1 cell proliferation by approximately 75%, 40%, and 57%, respectively. JQ1 appeared to be the most promising and potent drug for treatment of pNETs by enhancing the percentage of senescent BON1 cells and the percentage of BON1 cells in G1 stage of the cell cycle, and by significantly increasing the apoptosis by about 4.5-fold. No studies on these molecules are available on in vitro and/or in vivo models of *MEN1* mutated pNETs.

DNA hypermethylation characterized MEN1-mutated pNETs. Two DNA hypomethylating agents (decitabine and azacytidine) have been approved by the FDA for treatment of specific hematological malignancies; they could be studied as potential therapeutic options for MEN1-associated pNETs that show DNA hypermethylation.

ALT is a pro-oncogenic mechanism applied by a minority of cancers, alternative to the telomerase activation, to prevent the progressive shortening of telomeres during cell replication and avoid the senescence of proliferating tumor cells. ALT-positive cancers usually have a lower mean survival than non-ALT tumors and, in many cases, they appear unresponsive to standard therapies. Since ALT is a unique process that extends telomeres through exploitation of DNA repair machinery, it may present several druggable targets for treatment of ALT-positive tumors, including MEN1 pNETs, where ALT has been found in about 25% of cases. Potential ALT-related therapeutic targets and therapies that may be employed to exploit these new targets have recently been reviewed [53]. The telomere-specific FISH analysis of human tumor biopsies could help to determine the ALT status and identify patients that could benefit from an ALT-target pharmacological therapy.

Epigenetic-based targeted therapies represent a promising perspective for the medical treatment of MEN1 pNETs. However, several issues and challenges remain to be resolved on the use of these therapies, such as the specific targeting of tumor cells within the affected tissues, to achieve an effective, constant and lost-lasting effect on target cells, and reduce/avoid side effects on healthy cells. Prospective clinical trials on MEN1 pNETs are needed to translate epigenetic drug discovery to patients.

## Figures and Tables

**Figure 1 ijms-22-04041-f001:**
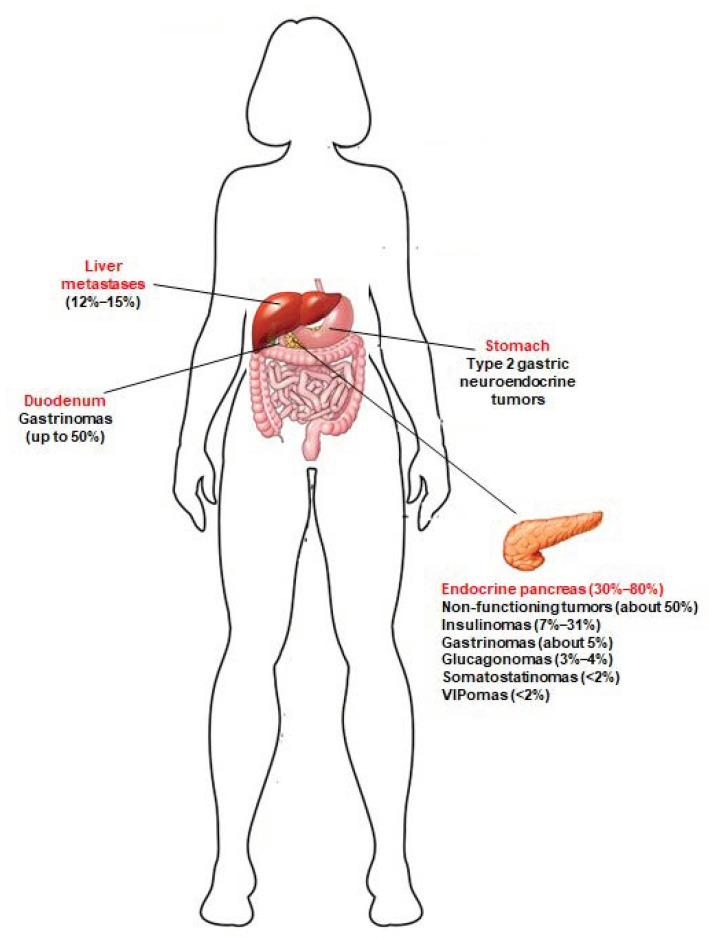
Neuroendocrine tumors of the gastro-entero-pancreatic tract associated with MEN1 syndrome. Prevalence of these tumors in MEN1 patients is indicated between brackets. Type 2 gastric neuroendocrine tumors develop in MEN1 patients, as a consequence of Zollinster–Ellison syndrome (ZES) from duodenal and/or pancreatic gastrinomas; to date, little is known about the true prevalence of these gastric neuroendocrine lesions in MEN1 patients. Jensen et al. [5] indicated a prevalence of 7–35%. Recently, Manoharan et al. [8] demonstrated that type 2 gastric neuroendocrine tumors occur only in MEN1 patients with ZES, and less frequently than previously reported (5.3% of the total of MEN1 patients and 12.5% of MEN1 patients with ZES).

**Figure 2 ijms-22-04041-f002:**
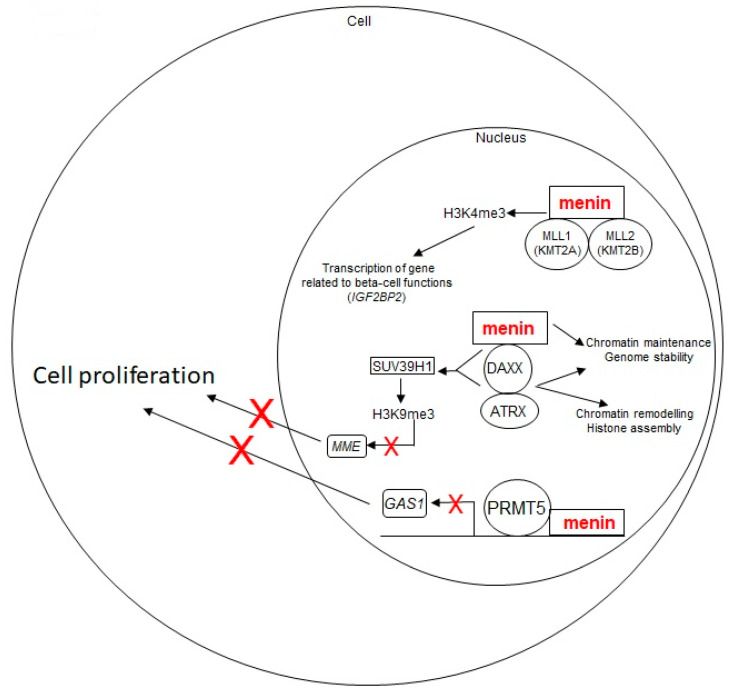
Schematic representation of menin-regulated pathways with a potential role in pNET tumorigenesis. Wild-type menin acts as a tumor suppressor for pNET development. It plays a role in chromatin remodeling and genomic stability and, via its direct interaction with the MLL1/MLL2 and DAXX-ATRX complexes, in histone methylation-driven gene transcription, and negatively regulates cell proliferation. Loss of wild-type menin results in an uncontrolled cell growth, leading to tumor development. The red X marks indicate the inhibition of a biological process (i.e., gene transcription, cell proliferation)

**Table 1 ijms-22-04041-t001:** Epigenetic mechanisms involved in MEN1-related pNET tumorigenesis.

Epigenetic Mechanism	Consequence of Loss of Menin Function	Effect	Role in MEN1 pNET Tumorigenesis
Histone 3 methylation	Loss of H3K4me3 and increase of H3K27me3	Downregulation of target gene expression mainly linked to beta cell endocrine function (*IGF2BP2*)	Development of functioning tumors of the beta cells of the pancreas
Histone 3 methylation	Loss of H3K9me3	Overexpression of *MME* gene	Increased and uncontrolled growth of neuroendocrine cells. Risk of liver metastases
Histone 4 methylation	Loss of the PRMT5-mediated H4R3me2	Expression of *GAS1* gene and subsequent activation of the Hedgehog signaling	Upregulation of tumor cell growth at beta islets
DNA methylation	Hypermethylation of CpG at *CASP8* gene promoter	Repression of *CASP8* gene expression	Prevention of apoptosis of tumor cells
DNA methylation	Hypermethylation of CpG at *RASSF1* gene promoter	Repression of *RASSF1* tumor suppressor gene expression	Loss of cell cycle arrest and promotion of cell growth
DNA methylation	Hypermethylation of CpG at *MGMT* gene promoter	Repression of *MGMT* gene expression	Loss of the protective action of the O-6-methylguanine-DNA- methyltransferase against exogenous alkylating agents, which are potent carcinogens

IGF2BP2 = Insulin-like Growth Factor 2 mRNA Binding Protein 2; MME = Membrane Metallo-Endopeptidase; PRMT5 = Protein Arginine Methyltransferase 5; GAS1 = Growth Arrest Specific 1; CASP8 = caspase 8; RASSF1 = Ras Association Domain Family Member 1; MGMT = O-6-Methylguanine-DNA Methyltransferase.

**Table 2 ijms-22-04041-t002:** Main treatments currently employed for MEN1 pNETs.

**Surgery**
**Pharmacological therapies** *Biotherapies* Long-lasting somatostatin analogues (octreotide and lanreotide) Inhibitors of mammalian target of rapamycin (everolimus) Inhibitors of tyrosine kinase receptors (sunitinib) *Chemotherapies* Alkylating agents (streptozocin, temozolomide, and cisplatin) Anti-microtubule agents (etoposide and docetaxel) Topoisomerase inhibitors (doxorubicin and irinotecan) Antimetabolites (5fluorouracil, capecitabine and gemcitabine) Cytotoxic antibiotics (actinomycin D, mitomycin C, doxorubicin and mixoxantrone) Combination of more than one chemotherapy compound
**Tumor targeted radiotherapy** Peptide receptor radionuclide therapy with somatostatin analogues (octreotide, octreotate, dototate, or dotatoc) bound to ^90^Yttrium or ^177^Lutetium

## Data Availability

Not applicable.

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
