# Peer review of "Pancreatic Neuroendocrine Neoplasms in Multiple Endocrine Neoplasia Type 1"

_ijms, 2021, doi:10.3390/ijms22084041_

Round 1
Reviewer 1 Report
This is a well-written review on a focused topic, pancreatic NETs in MEN1 syndrome. They touch on basic genetics, cell modeling and the future of therapeutics.
The sections are well written, the only suggestions I have is that the genetics section is dense, and would benefit from a figure on how the different proteins and signaling molecules interact (e.g. DAXX, PRMT5).
Similarly the therapies is densely written and could benefit from a table summarizing the text.
Author Response
Please find it attach a Word file with the point-to-point responses

Reviewer 2 Report
This is a review paper on pancreatic neuroendocrine neoplasm in MEN type 1. It is well-organized.
“Cancer” is used in the title. I think “neoplasm” would be better.
Many figures are put in the text. If some figures would be shown in tables, it would be easy to understand.
Regarding systemic treatments, they can be classified to some categories. They should also be shown in tables.
Author Response
Please find in attach a Word file with the point-to-point responses

Reviewer 3 Report
The manuscript by Francesca Marini and colleagues reviews pancreatic neuroendocrine tumors (pNETs) in Multiple Endocrine Neoplasia type 1 (MEN1).
This is a comprehensive, up-to-date and well written review of the topic. The relevant literature is cited and discussed. There are only a few minor comments.
I would suggest changing the title from “Pancreatic neuroendocrine cancer in Multiple Endocrine Neoplasia type 1” to “Pancreatic neuroendocrine neoplasms in Multiple Endocrine Neoplasia type 1”.
Line 250: “Moreover, the Authors [27] evaluated” – please change to “Moreover, Lin et al. [27] evaluated”
Line 392: “severe post-operative complications, such as exocrine gland insufficiency and diabetes”. Usually, postoperative complications are surgical/medical complications. I would rather call it severe post-operative sequelae or consequences.
Author Response

(The authors gave the same response as above.)
